# Presence of the Past: Digital Narrative of the Dennys Lascelles Concrete Wool Store; Geelong, Australia

**Md Mizanur Rashid \*** , **Chin Koi Khoo** , **Sofija Kaljevic and Surabhi Pancholi**

School of Architecture and Built Environment, Deakin University, 221 Burwood Hwy,
Burwood, VIC 3125, Australia; chin.khoo@deakin.edu.au (C.K.K.); sofija.kaljevic@deakin.edu.au (S.K.);
surabhi.pancholi@deakin.edu.au (S.P.)
\* Correspondence: md.rashid@deakin.edu.au

**Abstract:** Recreation of the past—of historical buildings—sits at the intersection of the spatio-temporal manifestation of cultural memories, socio-cultural meanings, values, and identity remolds, and refines the existing understanding and sense of place. Digital technologies have become a popular tool in recreation of the past by creating a new body of knowledge and historical discourse based on identifying the gaps within our written histories. Designers and policymakers around the world have been exploring various tools and technologies, such as diachronic modeling, yet there is a gap in evidence-based understanding regarding the actual functioning and success of applications for placemaking. This paper, therefore, sets out to scrutinize the role of digital technologies in facilitating digital placemaking. To do so, it investigates the potential of a new "digital heritage" narrative in the revival of the lost architectural narrative of the Dennys Lascelles wool store, Geelong. The proposed paper aims to investigate the potential of a new "digital heritage" narrative and storytelling as a means towards a digital placemaking framework. While exploring the new and unique capabilities provided by the digital narrative in capturing, simulating, and disseminating lost heritage, it will further imbue a sense of place by connecting the everyday city dweller.

**Keywords:** digital narrative; reconstruction; memory; placemaking



## 1. Introduction

### 1.1. Background

The meaning of a building in collective memory is intrinsically attached to the process by which it was produced, the manner it was experienced through time and space, as well as the way it was perceived. As memories are attached to the development of a sense of place and have attachment to its users, capturing the memories and the lost layers are crucial to preserve the built heritage of a place. In this regard, the city of Geelong posits an interesting case, as the city has changed rapidly, along with the mode of production and economic activities. Hence, the architecture of the city of Geelong in collective memory sits intrinsically at the intersection of multiple narratives as a palimpsest. It raises questions about gaps, or histories untold, and seeks what is borrowed and lost through time and space to reconstitute the heritage. The rise of the wool industry and the development of the port town of Geelong could be dated back as early as 1836 [1]. Since then, development and growth of the town has been closely connected with the wool industry for the next one and half a century. As wool became a booming industry for Geelong, the space for wool stores became premium and, hence, in the next few decades, collection of wool stores and warehouses became the main feature of Geelong's urban fabric. Major streets, such as Moorabool Street, Malop Street, Brougham Place, etc., in the Geelong's Central Business District (CBD), are studded with great varieties of new buildings related to the wool industry. The whole central city area was physically reshaped and redesigned to accommodate the new growing business, which had tremendous social and cultural impact on everyday life of the local community of that time.

The rapid rejuvenation of the city of Geelong in the late 1980s forced most of the industrial buildings to be relocated from the city center, which resulted in the heritage overlay of the city to (gradually) wipe out. The lasting legacy of wool making, as well as industrial architecture, including the legendary Bow Truss Building (Dennys Lascelles wool store), and many others, is eventually on the verge of getting lost from the memories of city dwellers, either because of demolition or major renovation. Tangible and intangible memories are being lost at the cost of development.

The integration of digital technologies for the interpretation of urban environment and spaces has opened up unprecedented possibilities in the way space can be experienced and into its meaningful interpretation. Recent studies have explored the integration of digital technologies in varied contexts, cultures, disciplines, and themes. Ranging from a discussion for their integration, aimed at creative, artistic, and cultural purposes [2,3], scholars have also called attention to their applications for better comprehension of human perception of public spaces [4–6]. These technologies have recently received global acknowledgement from designers and policymakers, for their role in increasing awareness on history and culture, as well as bringing together urban audiences around issues of public interest [7–10]. However, due to the elementary nature of this field, there still lies a big gap in terms of analysis of applications of newer inventions and applications. Secondly, the subject lies at the intersection of many disciplines, such as architecture, urban planning, urban informatics, neuroscience, and cognitive psychology. Despite the development of a number of theoretical frameworks in recent decades, which have attempted to interweave diverse perspectives together, it is yet to be explored from applied research based on an understanding of assimilating interdisciplinary perspectives.

To address this gap, the current research team recently completed a pilot project: "Digital narrative for placemaking: Diachronic Modeling of the Dennys Lascelles Concrete wool store; Geelong" to test and demonstrate the role and scope of digital technologies for the revival of the lost architecture of the Dennys Lascelles wool store, Geelong. Based on this project, the key research questions that this study aims to address are: (i) how can digital technologies, such as diachronic modeling, be applied as a tool for the facilitation of placemaking, in spaces and structures that hold heritage significance? (ii) How can we define a conceptual framework that aims to establish a connection between actual and digital recollection of memories represented through narratives and digital experiences?

### 1.2. The Case of Dennys Lascelles Wool Store

The Dennys Lascelles Concrete wool store, popularly known as Bow Truss Building, is an early 20th century industrial building with an expansive concrete roof; it once stood on Brougham Street, where the modern steel and glass Transport Accident Commission (TAC) building stands today. Along with the Barwon Sewer Aqueduct, this particular building is one of the two most celebrated engineering achievements by Edward Giles Stone, a civil engineer who pushed design boundaries with reinforced concrete in the early 1900s. It was claimed as being the largest flat-roof space in the world (almost an acre) without visible support; thus, creating a flood of natural light on the showroom tables by means of roof lighting [11].

The site was considered unique and was listed on several heritage registers, including the Register of National Estate and the National Trust register. The building was even nominated as a world heritage listing, which was supported by several international referees. Although this four-storied warehouse was not the first large reinforced concrete commercial building in Victoria, it is now the most (original) known, and possesses a number of unusual features. Its bowstring roof trusses span 182 feet and contribute a technological feat to the building and a strong visual element to the Geelong skyline. The external cladding is also of reinforced concrete and, thus, is structural, as well as decorative, in a simple Art Nouveau style. As part of a complex (in present day, the National Wool Museum), which began on its site in 1872 under C.J. Dennys, this building perpetuates the advances made in wool marketing by the firm in their earlier buildings.

Unfortunately, the rapid rejuvenation of the city of Geelong in the late 1980s forced most of the industrial buildings to be relocated from the city center. The heritage overlay of the city has been gradually wiped out due to economic pressure. In May 1990, the building was destroyed after the State Government of Victoria intervened to override the state's heritage body on its significance. The site was left as a car park for twenty years until the TAC building was constructed. The lasting legacy of wool making, as well as industrial architecture, including this legendary Bow Truss Building and many others, is on the verge of being lost from the memories of city dwellers. Tangible and intangible memories were wiped out at the cost of development. It is high time to respond to this situation.

*1.3. Recollection and Reconstruction of Memory using Digital Heritage*

As most of the buildings related to Geelong's booming wool industry, including our case, are demolished, its architectural impact on the morphology of the town is almost lost. Recollection and reconstruction of memories related to wool industry architectural objects and artefacts, and activities tied to it, varies, based on the physical presence of the building in question, because human memories are directly attached to the physicality of our environment. Consequently, the physical disappearance of the wool industry artefacts often means the disappearance of the memories related to it. At the beginning of the 20th century, Halbwachs [12], one of the most influential scholars of collective memory in social sciences, explains how all memories are recalled from the outside: the physical environment in which one resides offers different material means for memories to be reconstructed and recalled in the present. Thus, architecture has always been tied to the study of human memory.

In her work on medieval and renaissance memorial strategies, *The Art of Memory*, Yates [13] explains how mnemonic systems developed in ancient, medieval, and renaissance ages were based on the utilization of visualized human settings, where the placement of allegorical images happened inside imagined or recalled architectural edifices. She observes that this "method of loci" or "the art of memory", which uses loci and imagines, places, and images, was invented by the Greek poet Simonides and recorded in 264 B.C. for the first time. We know today that this memorization technique was used by Greek and Roman orators equally, the main sources being Ad Herennium of an unknown author, Cicero's De Oratore, and Quintilian's Institutio Oratoria. Yates [14] claims how this classical mnemonic device "consisted in fixing in memory as series of places, usually places in the building, or buildings, or in the streets of the city" (p. 573) was utilized to create spatial images and align them in a particular order where every one of them represents a specific point of the speech. "We have to think of the ancient orator as moving imagination through his memory building whilst he is making a speech," Yates [13] writes, "drawing from the memorized places the images he has placed on them" (p. 3).

In the intriguing essay about the interdependency between "built form and human thought," Parker [15] claims how memory, as our stock of knowledge, borrows the structure of its meaning–context from architectural and environmental works. The author argues, "It is evident that architectural imagery and structuring extend into intra-mental knowledge and memory" (p. 150), and continues how the reverse extension also occurs: "Memory, personal and social value and key societal concepts also extend into architecture and material culture, where they gain stability and easy accessibility" (p. 150). This mutual extension of architectural imagery into memory, and memory into architecture, causes our physical premises to become our conceptual premises, continuously co-creating each other. Parker [15] named this correlation between the two as architectonics, "the art of constructing systems of thought, and thus stable contexts for knowledge and ideas . . . when it can draw on aesthetically charged, concrete imagery" (p. 151). In other words, mental architectonics grounded on three-dimensional, visuo-spatial structures, can sustain greater permanence and organizational complexity in the mind of a user.

Similarly, Gestalt psychologist Rudolf Arnheim [16] recognizes this correspondence between human thought and architectural form. He claims that human thought has

always been "architectural" in its nature. Analysis process, development of an argument, recollection of a memory, storytelling—all of them need to (first) be laid out spatially. Arnheim argues: "When the human mind organizes a body of thought, it does so almost inevitably in terms of spatial imagery . . . any organization of thought assumes the form of architectural structure" (p. 271). As such, the mind is not only reinforced, but it relies on visualized architectural images as a part of ongoing mental processes.

Consequently, the connection between human cognition processes and visualization of images of the built environment is unquestionable. In *The City of Collective Memory*, Boyer [17] writes: "Architecture and city monuments can become artefacts and traces that connect the past with the present in imaginative and inventive ways and help to build a sense of community, culture, and nation" (p. 309). As such, the value of a finalized physical building is not limited to its immediate purpose—to protect and organize our lives in a creative, effective, and comfortable manner—but its significance lies in an array of intangible meanings that are assigned, through time, to its physical existence, and people's interactions with it as a part of the broader urban landscape. Some would argue the immaterial aspect of architecture—that only occurs after the building assumes its physicality—represents the greatest value that a material edifice can attain, because it aids the human thought process and recollection, and has been a witness to people's lives and histories throughout time.

Multimodal representations of space and its role in shaping the experience of place, and development of sense of place, has been the focus of early classic works by renowned scholars [18–20] Watkins [21] divides the experience of space as representations of space (conceived space), spatial practices (perceived space), and representational spaces (lived space). In his classic works, Lefebvre's ideas of spatial triad brings together the materiality, representations, and imaginations to be intertwined in the simultaneous production of space. These works contend for an evident and strong interconnection between the way spaces are represented, and the actual realities attached to places. In addition to this, scholars from Relph [18] to Duranti [22] have highlighted the inter-subjectivity of experience of space where one's sense of place can be shaped and re-shaped through the images and experiences through various mediums. To understand the concept of multi-modality better in contemporary terms, it is critical to look at representation of spaces through multi-layered and diverse digital mediums. Deconstructing the classic ideas of Relph [18]—that highlighted communally constructed identity of place—and inter-twining them with the modern application of digital technology, Croy and Wheeler asserted its role in the formation of collective and personal image of place. Henceforth, the integration of technologies has blurred the boundaries between the experiences of inter-related aspects of space. Eventually, it enriches it further by extending it from single-dimensional representations through pictures, images, or artefacts to immersive experiences through modern, digital mediums.

Drawing on the latest discoveries in the fields of neuroscience and cognitive psychology, Goldhagen [23] observes that the physical environment that we inhabit during a particular experience plays the dominant role in memory itself. These findings are hardly novice, as they represent conformity and an addition to many studies that have been conducted in the last century throughout different scholarly fields on the relation of place and memory. Goldhagen explains: "In the contemporary world, where our environments are overwhelmingly built environments, what this means is that the buildings, landscapes, and urban areas we inhabit are central to the constitution of our autobiographical memories, and therefore to our sense of identity" (p. 83). Neurologically, different kinds of long-term autobiographical memories are consolidated and prepared for long-term storage in the part of the brain called the hippocampus and the adjacent parahippocampal region. Working with the other areas of the brain, this part of the brain also facilitates our ability to navigate space. "Place cells," Goldhagen claims, "enable us to both identify a place and consolidate a long-term memory" (p. 84). Therefore, the cognitive processes of autobiographical memory situate the built environment inside us, and that way, it constitutes "the internal

architecture of our lives" (p. 88). Therefore, visualized place-based experiences create a unique framework for self-understanding and perception of who we are.

Based on the premise that architecture could be a unique and revealing frame of inquiry to gain insight into human nature, attitudes, values, worldviews, and immaterial/material culture, the main aim of the project is to address three key issues using a digital heritage narrative as a tool for reconstruction and recollection of the past of Geelong's wool industry. With the aim of improving the access of general audiences to the architectural–archaeological heritage content of Geelong's history, the project aims to deliver two main tangible outcomes:

1.  A digital model visualizing the lost architectural asset of Dennys Lascelles wool store based on the fragmented archival data, available for exploring the storyboard instead of a photo realization of the original building. The key idea is to initiate and demonstrate the process of understanding how digital placemaking can complement physical placemaking, and vice-versa, through utilization and reconstruction of collected heritage material.
2.  To explore and establish a new conceptual framework that will define successful relationship between (a) recollection and (collective) memories and multiple narratives of the place, and (b) creating a web-based framework for bottom up user-based and user-driven storytelling online immersive environments, which can further be reapplied in the heritage industry with the purpose of meaningful, ethical, and community driven representation of the past. This section may be divided by subheadings. It should provide a concise and precise description of the experimental results, their interpretation, as well as the experimental conclusions that can be drawn.

## 2. Materials and Methods

### 2.1. The Idea of Diachronic Modeling for Digital Heritage

The proposed Diachronic or 4D modeling is a method that is based on the premise that consider architecture as a process rather than product, due to different internal and external conditions, and transforms through time. It relies hugely on a linked open database (LOD), which is a linked database for data collected for a particular site from the earliest known date. This LOD usually collates all the relevant information in a scientific way and tries to fill in the lacuna using scientific/architectural reasoning by observing other social, political, cultural, and economic, etc., conditions. Based on this analysis, it proposes a possible three-dimensional transformation of the site in four-dimensional coordinates (X, Y, Z, and T) that is also linked with particular conditions both diachronically and synchronically. It places a building in the crossroads of multiple historic narratives both tangible and intangible. This has not been used in capturing architectural history and heritage in Australia to date. The novelty lies in applying the visualization capabilities provided by developments in digital design to make clear the multiple aspects of heritage buildings, particularly the aspect of firm relations between architecture physicality and memory for the purpose of recording and disseminating of architectural heritage—showcasing how the building was actually conceived, constructed, and used—and in so doing, unravelling multiple historical layers and connection to a broader geographical and cultural domain.

These dynamic narratives of the sites (with the end dimensional hybrid capturing techniques) in Geelong's CBD, depicting the changing forms and patterns of storytelling of this industrial town over time, and related official historical narratives, is an innovative and a novel research outcome, which will serve to reconstruct the lost heritage building to define its place in the collective conscience of the present day Geelong Community and beyond in a tangible and accessible format.

### 2.2. Collection and Analysis of Historical Data

The main challenge at the beginning of this research was the limited visual information and reference, such as floor plans and sectional drawings to help understand the spatial position and architectural structure of the demolished Dennys Lascelles Austin Bow Truss

wool store. Absence of the construction documentation also means it was practically impossible to rebuild all engineering details on the computer. Therefore, it was decided to make the 3D model match as close as possible with the collected sets of textual and visual reference sources from newspaper archives and old photographs. Fortunately, some former studies around this demolished wool store, documented by architectural historian Miles Lewis [24], provided valuable insights into the construction at the time of how the giant concrete bow trusses pass through sawtooth roof structures in order to welcome abundant natural light. Some images particularly captured the key features of the building, including facades, roof panels, truss pairs, northeast entrance, and the demolition photos, revealing the internal structure of the building. Moreover, both the TAC building at the site and the remaining adjacent heritage architecture, now known as the National Wool Museum, also offer useful reference to speculate dimensions for the Bow Truss wool store building. At this stage, there was no continuous narrative of the visual aspects of the building, what we had at this moment of time is fragments of images of different parts of the building. Hence, we took an approach of iterative modeling with the data in hand. Initially we prepared a set of measured drawings of the building using images and reverse calculating the dimensions using human scale as reference. These initial drawings were used as baseline data for modeling. Later, newer data were continuously collected, managed, and analyzed to be served as historical reference inputs for the iterative 3D modeling in Rhino.

### 2.3. D Modeling of the Lost Bow Truss Wool Store Building

Rhinoceros 3D (Rhino) was chosen as the modeling tool for digital reconstruction because of its precision and sophisticated control of geometric shapes during the redrawing process, which is suitable for the condition of continuously updated reference data. The main component that characterizes this wool store is its massive, column free top floor, which measured approximately 55.5 m × 52 m. By using the measuring tools in Rhino, such as Snaps, Evaluate Point, Measure Distance, and Angles, the spatial relationship between each chord, rod, and roof line could all be accurately calculated based on the bitmap of the girder section (Figure 1). Once the modeling of the first pair of bow truss was completed, it then could be duplicated and positioned at the correct place by using the tool of Array in Rhino (Figure 2). Similarly, many other repetitive components were following this modeling strategy: the windows, roof panels, facade columns, and external decoration (Figure 3). It was important to bear in mind that all different single pieces were connected with each other, if one component of the building were changed; then everything else should be updated accordingly. Therefore, all supportive lines, surfaces, and geometry could be saved in one layer as back up in case earlier modeled elements were needed, and all these digital records could be considered as internal data reflecting the 3D manufacturing and testing in the process of digital reconstruction (Figure 4).

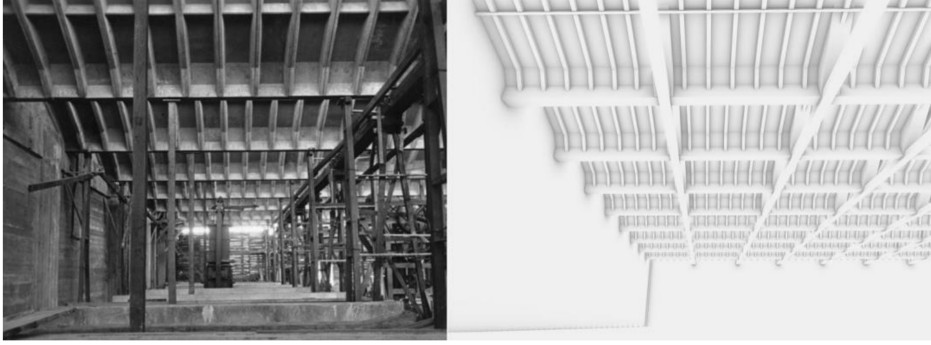

**Figure 1.** The photo of the interior (**left**) serves as the reference to reveal the roof and ceiling structural components, represented through the digital reconstructed model (**right**).

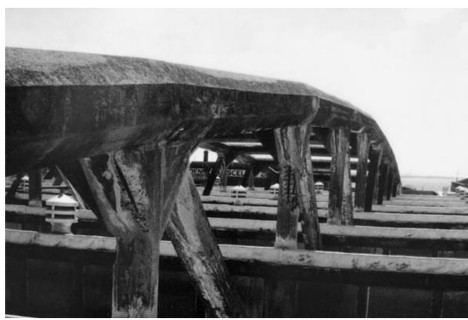 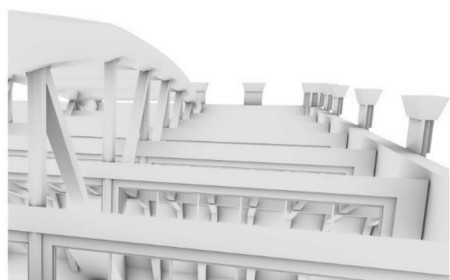

**Figure 2.** Left: the photo of the bow truss structure above the saw-tooth roof system. Right: the digital reconstructed model of the bow truss roof is modeled based on the reference of the previous photo.

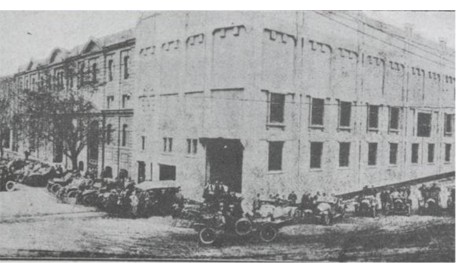 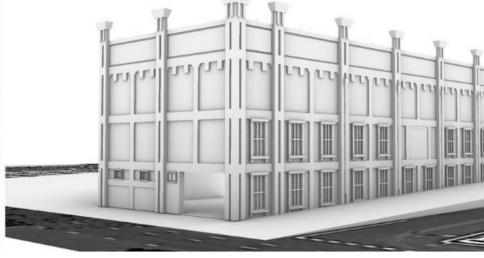

**Figure 3.** The north facade and entrance of the lost wool store (**left**) as the reference for the overall exterior features and elements of the facade (**right**).

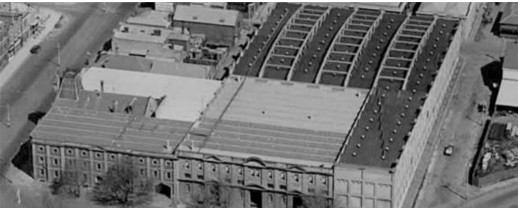 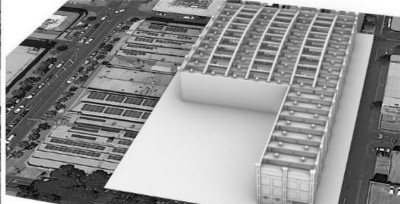

**Figure 4.** The bird-eye view photo (**left**) as the reference for the overall roof structure and facade of the digital reconstruction (**right**).

*2.4. Representation of the Architectural Narrative*

It was easy to navigate the reconstructed 3D model in Rhino, as one can zoom in, zoom out, and rotate the camera at any viewpoint to experience the Bow Truss wool store building (Figure 4). However, the architectural narratives were presented on a website, as a virtual exhibition to the public; therefore, the interpretation of the 3D model had to take into consideration which graphic works match up to the general audience. Two 3D images were made via the initial diachronic digital model to maintain the quality of the aesthetic and accessibility of the architectural narrative. One was an exploded isometric illustration of how this lost heritage building was constituted by sets of different components (Figure 5). These components were confidently produced using the initial measured drawing and old images. The other one was a series of cross-sections that explained the relationship between the Bow Truss structures and the whole building (Figure 6). Some images were taken underneath the model of saw-tooth window frames and roof panels, which complemented the missing visual narratives from the historical data set. Other representational images compared the lost Bow Truss wool store building with the existing geometric volume of the TAC building and National Wool Museum, showing the lasting architectural legacy within the urban context (Figure 7).

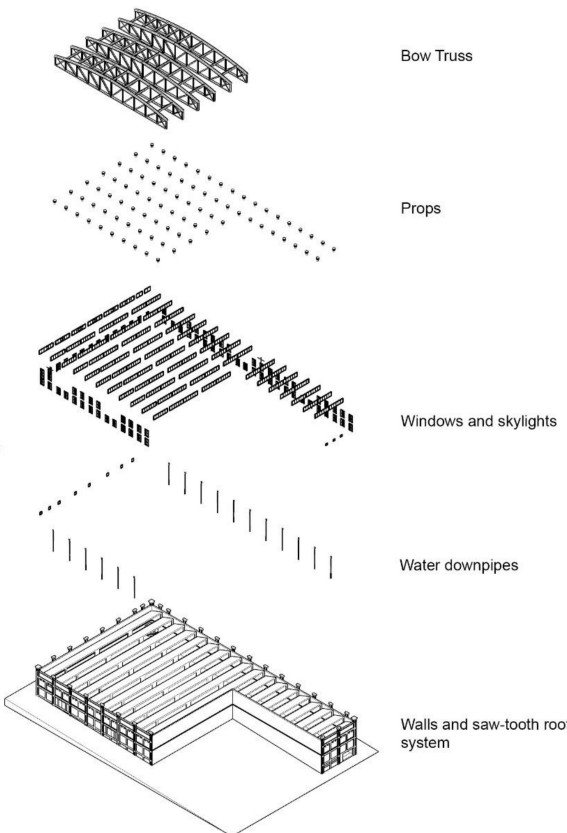

**Figure 5.** Exploded isometric view illustrated the various components to form the overall Bow Truss wool store building.

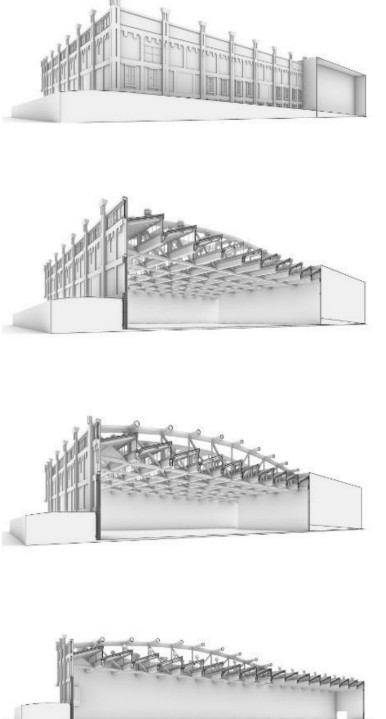

**Figure 6.** The sequential sectional perspectives to demonstrate the roof structure that consist of the various components include the reinforced concrete bowstring trusses and saw-tooth roof system.

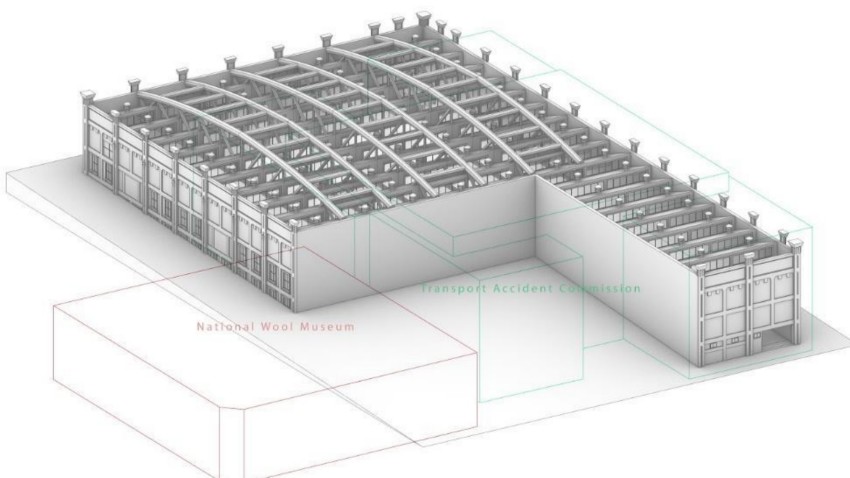

**Figure 7.** The three-dimensional (3D) visual comparison of the geometric volume between the existing National Wool Museum, Transport Accident Commission (TAC) building and the lost Bow Truss building.

Figure 8 illustrates the additional information of the various isometric views that interpret from different directions to provide the comparative insight between the past and present urban context of the wool store. To understand the site and surrounding and its transformation in the last one hundred years we studied the archival/ historic aerial images of the CBD of Geelong and tried to trace how the site and the surrounding of the wool store have changed, along with changes of the urban fabric and infrastructure (railway), as shown in Figure 9. This narrative could be further elaborated as a dynamic diachronic model in the next level of the study.

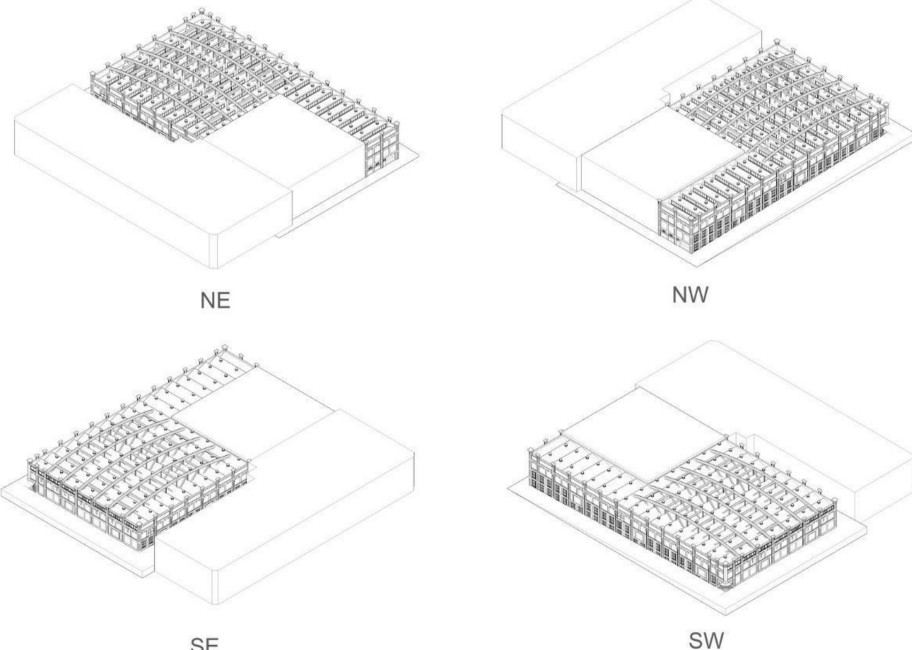

**Figure 8.** The various isometric views of the Bow Truss building: northeast, northwest, southeast, and southwest.

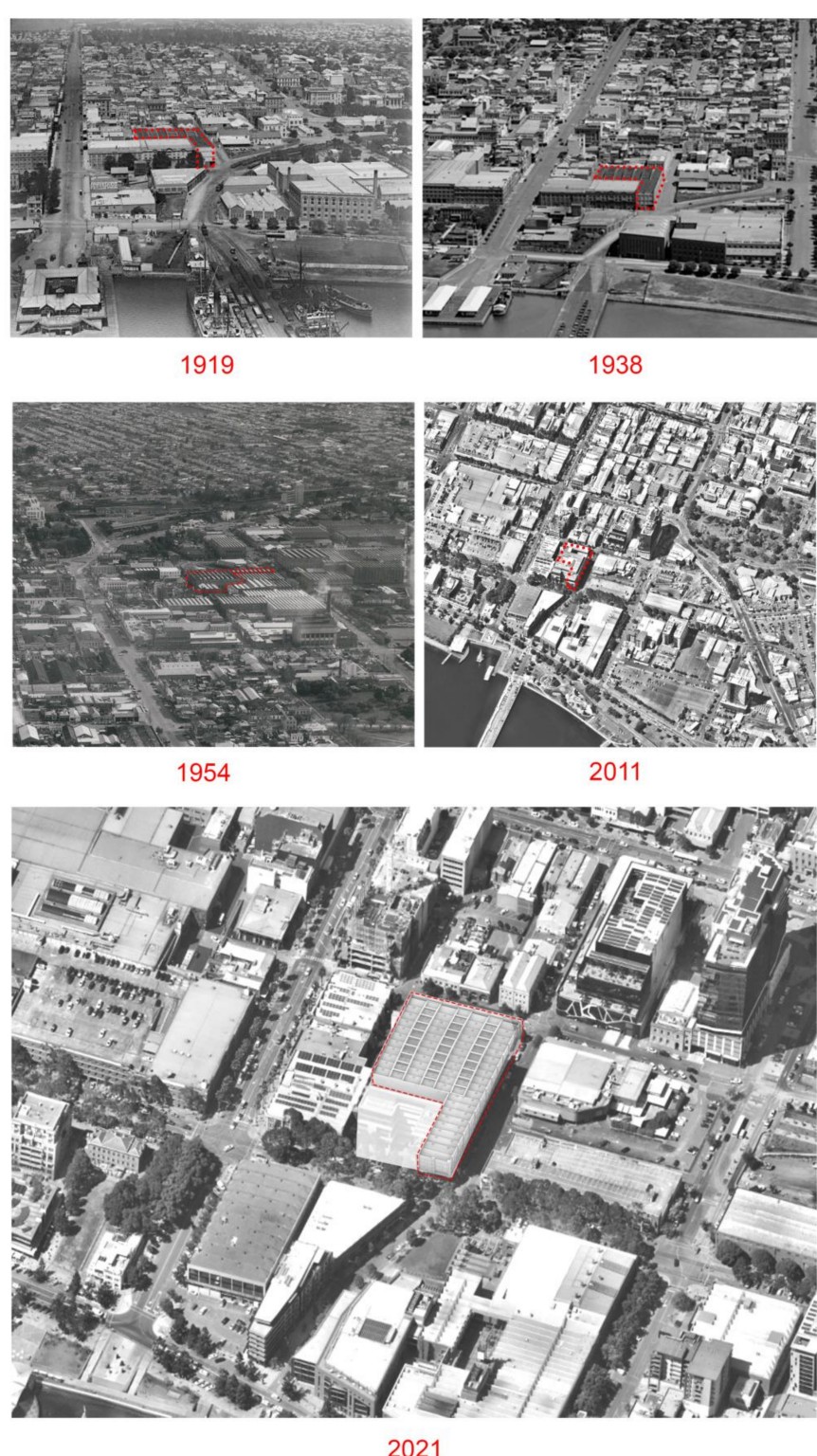

**Figure 9.** Aerial images showing the urban transformation in the last one century around the site.

## 3. Result

The outcome of the project reemphasizes that the meaning of the building and associated collective memories are intrinsically embedded in the process through which the building was conceived, produced, and endured a series of socio-cultural changes [25]. Hence, the history of our built heritage should be centered on the understanding of human experiences, history, and narratives through historic fabric, structures, and remains [26].

Being in a telescopic distance, it is always difficult for present day users to be immersed in the past, embodying the memories and meaning of a building, without participating in the performance of rituals and social acts [27]. Buildings of the past, with their very existence, are attached to collective memories of certain groups [28]. Thus, architecture becomes the most tangible and durable object of remembrance, though it could be paradoxical and contested when it engages with collective memory [29].

Virtual heritage has significant implications on non-invasive restoration and preservation of the monuments. It generally provides an immersive multimedia experience through a computer-simulated environment that can simulate physical presence in places in the real world [30]. It further provides scope for an interdisciplinary research environment by developing a rich database of the digital assets for the conservators, historians, and archaeologists, to restore the historical sites, as well as heritage preservation. Although, for most of the cases, the 3D virtual models contain accurate data and help with restoration, whether they could capture the associated meanings and memories, especially the intangible values, are a big question for an architectural historian. Thus, the concept of building as "place" along with 3D articulation of the lost building comes in front. "Place" through the articulation of space has been at the concern of architectural theory and practice for the last few decades, and a widely discussed topic of architectural history and theory. Place can be understood in relation to space, though it is not created by mere three dimensionality of space. It is rather 'about the practices and politics of place and identity formation—the slippery ways in which who we are becomes wrapped up with where we are.' [31]. While the assumption of 'place', in the context of this chapter, is a meaningful interaction to a space where the user, environment, and the memory 'tell it's past . . . [and] contains it like the lines of a hand' [32], the 3D model of architectural space generally addresses the metric expression of form, shape, and material physicality.

The discipline of history and theory of architecture traditionally focus on the aspects of visual culture best represented by the most advanced image reproduction technique of that time [33]. Introduction of virtual reality (VR) and augmented reality (AR) certainly transformed our capacity to understand structures and resolve issues of plausible historic design that no longer exist [34]. With the incorporation of digital humanities into the mainstream research and dissemination process of architectural history, virtual imagery would certainly dominate the entire architectural realm. However, the question remains, to what degree (and how) virtual imagery should be used to convey meanings and memories, as well as interpret them correctly, particularly due to the heavy reliance and ocular-centric nature of these technologies. While these virtual realities may allow us to investigate and recreate the lost architecture, " . . . they are not likely to help us experience inhabiting that place, moving through that place, or understanding the dynamic and ever-changing relationship of people and place." [35]. The reason may be the overemphasis on the fidelity of the created objects and understanding of ocular engagement in fully understanding "place". Imagery has always been considered as factual and is used to provide evidence in " . . . legal cases and in science, photographs operate within the modality of actuality. The photographs [the visual] are meant to allow us to discern what actually occurred" [31]; however, it could be misleading if we put them in the wrong context. This is no different to evidence of the historical facts. Therefore, when utilizing visual technologies, such as VR and AR to portray architectural history, a place should be understood to its fullest within the context. We know that icons and symbols provide meaning to architectural form, and these meanings are the intangible aspects of the heritage. As Pallasmaa argued, "... technological culture has ordered and separated the senses... Vision and hearing are now the privileged sociable senses, whereas the other three are considered as archaic sensory remnants with a merely private function, and they are usually suppressed by the code of culture" [36]. This suppression of other senses and the ocular facilities might provide a false sense of place by conveying a different meaning or sometimes creating a new one. By just creating an ocular narrative of a place through virtual and augmented realities, we could ultimately be removing the intangible feelings, emotions, and cultural

memories attached to a space. Without the true encompassing narrative " . . . no matter how indexical, suitable, or numerous the representations of an object are, what is on the screen will always resolutely remain a representation that stands in for something else" [33]. This re-production of a space becomes its own entity and "establishes their own versions of the past" [37].

Hence, in this particular project, visual aspects of the reconstruction, i.e., photo realization, is considered as part of the narrative of the building. It is considered as just the beginning of disseminating heritage value and connecting the user rather than the end product. As discussed earlier, architectural heritage is something more than the physical form. A building is a place for doing different activities in and around. To understand the architecture of this monument, mere virtual reconstruction of the three-dimensional form would not be sufficient. In order to create a virtual environment embodying the essence of place is inevitable. Usually the role of "place" is a virtual environment as a locator of objects [38]. Thus, the architectural heritage in the collective memory sits intrinsically at the intersection of multiple narratives as palimpsest. Hence, to recover the memories of this building, it is required to identify and examine these narratives along with the virtual modeling. The issue of 'place' becomes crucial while reconstructing the past with limited resources in hand, which are fragmentary and inconspicuous in nature. The temporal distance and the lack of understanding between photo realization of the actual architecture and creating an unbiased sense of place remains at the crux of the problem.

From that aspect, the digital narrative of the Bow Truss wool store building opened up new opportunities to the public, to communicate and interact with the state's heritage significance. The same approach with the 3D model can also be applied to bring back other damaged, unbuilt, or demolished buildings. For future work, it is planned to scale up the project to use a case study research strategy and the content of Geelong's wool industry heritage to empirically investigate how to communicate architectural buildings with different social values and collective memory (of renovated vs. demolished buildings) among the local community.

## 4. Discussion

As this particular project had limitations due to the Corona Virus Pandemic (COVID-19) situation, in terms of budget and accessibility, initial aims and objectives were amended later to frame it within the capacity of the project team.

Considering the significant changes of the situation—limitation of budget and limitation of time—the research team decided that, instead of providing an onsite VR/AR experience for public interaction, the amended proposal would aim towards developing a bottom up user based framework for capturing the narrative of the building. Aligned with the changed research focus due to existing limitations of physical movements in public space, the study explored in-depth variations of cross-media storytelling in relation to digital (online) placemaking, in order to understand how different digital/virtual platforms can diversify and expand heritage experience. For this purpose, the project was designed/redesigned/undertaken in the following way with five specific stages illustrated in a flow diagram (Figure 10).

Stage 1: collection and organization of insufficient heritage data, both tangible and intangible. In this stage, a thorough search was made by the team into all the available digital archives in Australia. Using TROVE (an online database by Australian National Library) as the starting point, the team browsed through different national, state level, and local databases for images, maps, publications, newspaper articles, and any other relevant information regarding the building itself, city of Geelong, and the wool industry in the region. The idea was to collect any fragments of information that might have some kind of connection with the case study building. Hence, a wider and broader search was done to understand the context under which the building was incepted, constructed, and eventually demolished. Since the building was demolished, there was very little visual data available in different online databases. No drawings of the building were found.

Major descriptions about the buildings were mainly found in different newspaper articles between 1910 and 1917, describing this awe-inspiring building as an urban landmark. This fragmented information was initially organized chronologically using the online software Sutori, as an online interactive digital platform. Due to the lockdown in Victoria, Australia, the research team members could not meet physically and, hence, Sutori was a very good platform to interact with the information collected by different team members and edit if necessary.

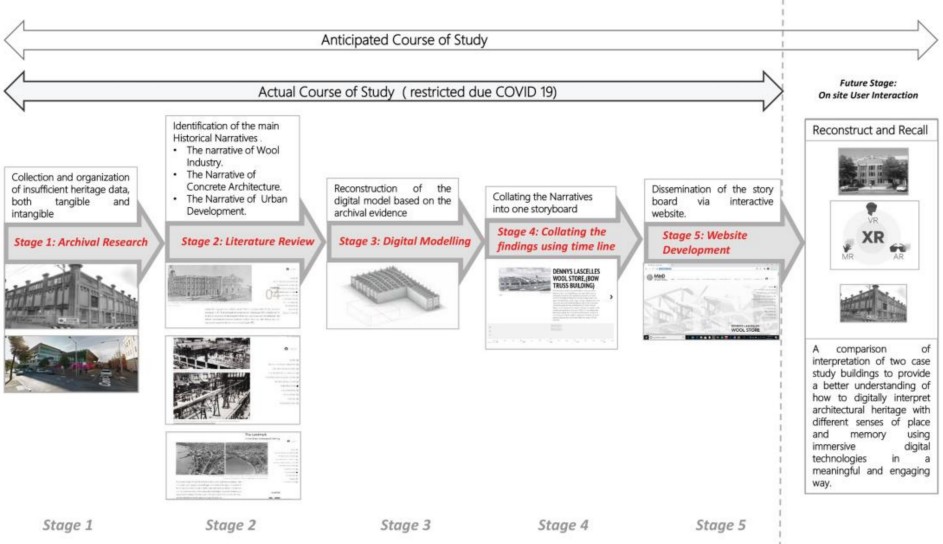

**Figure 10.** Flow diagram showing the anticipated and actual study of the specific five stages and the proposed future stage.

Stage 2: identification of the main historical narratives. Once the collected data were initially organized chronologically, the team focused on identifying different historical narratives associated with building. The team identified three major historical narratives: the narrative of wool industry in Geelong region, the narrative of concrete architecture, and the narrative of urban development in Geelong. The initial database in Sutori was then collated according to the three narratives and different research team members were assigned to look into different narratives, focusing on creating a storyline for dissemination.

Stage 3: reconstruction of the digital model based on the archival evidence. Once the draft storyline of the three narratives were created and a case study building was placed on the intersection of the three narratives, all of the information relevant to the building and its architecture was collated, and a 3D virtual model of the building and the site was made based on the available date. The detailed process of the model making, based on the fragmented resources, was described in an earlier section.

Stage 4: collating the narratives into one storyboard. This stage involved collating all three narratives and the virtual model of the building into one storyboard for the general user. The storyboard was designed in a simple and easily accessible way, avoiding all the research related jargon so everyday users could easily grasp the content. However, information of a more complicated nature was linked in such a way that whoever is interested, could also have easy access.

Stage 5: dissemination of the storyboard via an interactive website. The final stage of the research involved developing a bottom up user based web framework for capturing the narrative of the building, available at www.dennyslascelles.net (Figure 11). This website is, at this moment, open for user feedback and comments, and contribution through interactive forums. Any user who has memories associated with this building, as well as any images, drawings, or photographs that are relevant, are encouraged to share them through the website. It is anticipated that, after one year of running the website, feedback

and contributions will be collated with the main storyline. Hence, a web portal will work in both ways.

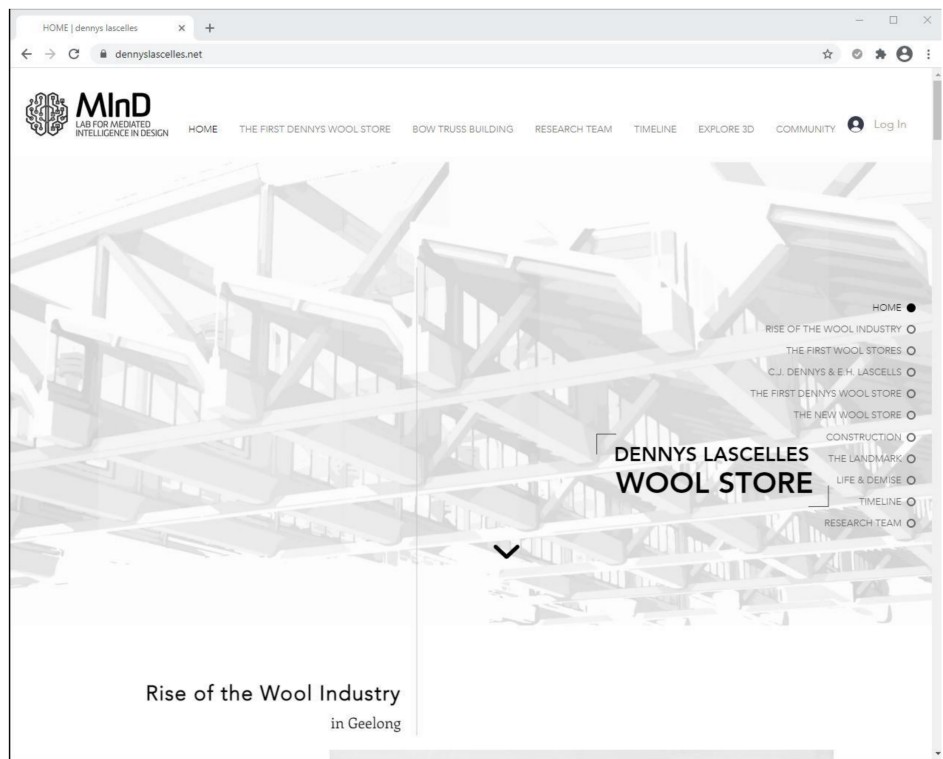

**Figure 11.** The bottom up user based interactive website for the project.

After completion of the project, we learnt what it meant to detach the heritage material from the physical location of origin, and consequently, how it could create a novel "digital heritage space" for the user. Throughout this process, the attempt was to understand the advantages and drawbacks of construction of "virtual" versus "physical" space in terms of representation of heritage material, and to understand how both can be utilized in a more creative and efficient way. Hence, the team has focused more on developing the digital model for exploring the storyboard instead of a photo realization of the original building. The key idea was to initiate and demonstrate the process of understanding how digital placemaking can complement physical placemaking, and vice-versa, through utilization and reconstruction of collected heritage material. In other words, the aim was to open the discussion, omitted so far, on construction of memory when the participant's experience was obtained through engagement with "virtual historical space" compared to traditional means of heritage representation related to specific physical space (traditional museums, galleries, monuments, etc.).

The project made valuable scientific contribution in two aspects. First, it clearly demonstrated that such virtual models could create better narratives of the building and town, which cannot be gained from old pictures and historic material, only those that are currently fragmented. Collation of these fragmented materials into a linked open dataset (LOD) that was developed in the early stage of research using Sutori is an open-ended database that could include newer materials in the future, which could be used later for further study or modeling. The second scientific contribution is the user-based bottom up dissemination method, using the website to include user feedback for the next stage of the research (the 4D interactive narrative). It was mentioned earlier that, due to constraints imposed by the pandemic, the scope of the study was curtailed short. However, this study strongly substantiate the idea of a 4D narrative to fully capture the lost architecture of the building and equally emphasize the need of using different remote sensing tools for that exploration.

Learning from the project necessitates a novel digital heritage interpretation approach [39] underpinned with the architectural theory, and supported by human-centered design and participatory design processes. That could be explored in the future by up-scaling the project for the potential use of cutting-edge immersive extended reality (EX) technologies, such as virtual reality (VR) and augmented reality (AR). The heritage content in XR will be relocated on a digital humanities geographic information system (GIS) platform, providing the general public an engaging experience through time and space as a 4D (3D + time) narrative. The results could be achieved through an interdisciplinary collaboration of researchers from architectural, design, heritage, and engineering areas.

In a situation where the heritage building is partially or fully lost, the project demonstrates that major interaction with the user is necessary for the experience to be memorable for the participant, which we tried to achieve through "digital storytelling" and creating new "digital narratives." The literature and research findings from the field of memory studies show that memory of a place and, consequently, place attachment and place identity, can be severely endangered, and even fully erased through time when the physical artifact, including an architectural edifice, disappears/ is destroyed. There are no empirical studies reports published so far on how memory construction and reconstruction occur through engagement with virtual heritage places. Whether it could be done by VR or AR application or through mixed reality is a matter to be investigated further. The finding of the project raises questions about that. As we know, the VR environment and real physical environment, by using AR, often has very different approaches because of the different natures of experience and human–environment interaction. VR provides a total immersion, often of places that are hard to access or do not exist (anymore). Such experience can be accessed from anywhere, anytime, using a head-mounted display (e.g., Oculus Quest). During the global COVID-19 pandemic in 2020, many museums created so-called "museums from home" experiences, including in VR, for visitors who were not able to access physical museum exhibitions [40]. AR, on the other side, augments physical reality by overlaying 2D or 3D information. The image is then projected on a smartphone, tablet (e.g., Pokémon GO game), or on special glasses (e.g., Hololens or MagicLeap). Current smartphones and tablets enable AR content to be accessed in situ, which is widely used in archaeology for interacting with 3D reconstructions of unpreserved heritage directly at the sites.

Despite the wide use of VR and AR in the heritage field, it is not well known how those technologies can support the representation of built heritage from the recent past, in regards to human engagement, with a place and place experience, particularly how those experiences impact memory construction, place attachment, and construction of identity related to a place. Moreover, how the same technologies can aid us in a process of successful placemaking.

Hence, as the next step, the team plans to take two wool industry heritage buildings in Geelong CBD as case studies to compare the quality of experiences of virtual versus physical engagement. One, the current Dennys Lascelles wool store that was eventually lost through time, and the other, the renovated building of the Dalgety and Co. wool store, which is currently used as the School of Architecture building for Deakin University. Through the planned empirical, qualitative comparative study of the two buildings, one reconstructed through various virtual technologies and the other still physically present on the location, the team will try to understand similarities and differences in human engagement between two places and, furthermore, how memories are constructed and reproduced in relation to each architectural structure.

## 5. Conclusions

The findings of this project further re-emphasize the need to explore in-depth variations of cross-media storytelling in relation to virtual (online) and physical (on-site) placemaking by application of different digital technologies, in order to understand how different digital/virtual platforms can diversify and expand heritage experience. It neces-

sitates learning what it means to detach/remove the heritage material from the physical location of origin and, consequently, can create a new "digital heritage space" for recalling experience of the placemaking and a sense of place.

In terms of disciplinary contribution, this research will provide significant contribution to re-center our position to look into architecture, engineering, history, and society from a different perspective, as listed below.

(1) It will provide opportunities to revisit the ways of seeing the past through software or the design of the interface.
(2) It will sustain the information through a linked open database (LOD) for further research
(3) It will help us understand the spatial aspects of the built environment through a digital medium.
(4) It will test multimodal forms of collecting, storing, and disseminating research data.
(5) It will raise a new research question: does virtual immersion presupposes another way of interrogation?
(6) It will generate new research knowledge from a paradigm of static databases to a dynamic and interactive field using an immersive environment.

The main issues outlining the scope for recall and reconstruction, in this case, concerns the potential of a digital heritage narrative as a means towards placemaking. While exploring the new and unique capabilities provided by the digital narrative in capturing, simulating, and disseminating 'lost' architectural heritage, it will further imbue a sense of place by developing a sense of pride, belonging to the everyday city dweller. It is not only important to understand the past, but also to predict/design the future of the city as well.

In addition to contributing significantly to academia (in terms of knowledge generation), this project, and the discussed method of dissemination, has the potential to make significant social benefits—long-term benefits pertaining to the heritage management of these fragile sites with the use of diachronic end dimensional capturing techniques, to enable this knowledge to be conveyed to both academic and targeted audiences.

**Author Contributions:** Conceptualization, M.M.R.; methodology, M.M.R., C.K.K., S.K., S.P.; software, C.K.K.; validation, C.K.K., S.K., S.P. and M.M.R.; formal analysis, C.K.K.; investigation, M.M.R., S.K., S.P.; resources, M.M.R., C.K.K., S.P.; data curation, K.S; writing—original draft preparation, M.M.R.; writing—review and editing, C.K.K., S.K., S.P.; visualization, C.K.K.; supervision, M.M.R.; project admin-istration, M.M.R.; funding acquisition, M.M.R. All authors have read and agreed to the published version of the manuscript.

**Funding:** This research was funded by the School of Engineering and Built Environment, Deakin University Internal Funding (SEBE SIF), Deakin University 2020. The idea was for research to be aligned with the objective of MInD Lab (Mediated Intelligence in Design).

**Data Availability Statement:** The outcome of the research is available at the published website https://www.dennyslascelles.net/ (accessed on 17 December 2020).

**Acknowledgments:** We acknowledge Chun Wang for his support to do the modeling of the lost building and Raihan Rafiq for developing the website.

**Conflicts of Interest:** There is no apparent conflict of interest.

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
