# Peer review of "Presence of the Past: Digital Narrative of the Dennys Lascelles Concrete Wool Store; Geelong, Australia"

_remotesensing, doi:10.3390/rs13071395_

Round 1

Reviewer 1 Report

The paper presents a case study of creating digital narrative based on diachroic modelling of the Bow Truss building in Geelong. The paper is well organized, written and easy to follow. 

Then why have I suggested rejection? Because I can't see how this paper fits into the scope of Remote Sensing Journal which is "science and application of remote sensing technology". The authors recreated the 3D model of Bow Truss Building basing on photos and documents, so no remote sensing technology was used, the digital representation was modelled in Rhino.

As the paper is really nice and the work worth publishing, I would suggest submitting it again to other journal with more "digital heritage" - oriented scope.

Author Response

This particular issue of the journal of Remote Sensing is focusing on the use of virtual modelling in preserving cultural heritage. This paper highlights the issue of a building that is currently lost and only fragmentary information are available. The paper thus argues the necessity of Geo-Spatial platform (GIS) to retrieve the memory of the lost building in addition to just using virtual reconstruction. (Page 2; Line 64-68)

Reviewer 2 Report

An interesting paper! It would be interesting to have more details on the 3D modelling process, for instance accessing the level of confidence of the different reconstructed parts of the historical buildings. Are all the parts tridimensional reconstructed with the same level of confidence?

It would also be interesting to have a time-lapse regarding the historical evolution of the building over time and relate this with the collated digital narrative. It would potentially improve the "digital heritage narrative" and story-telling.

Author Response

Reviewer2:

  1. How does the paper could be more specific in terms of scientific contribution?

This issue has been addressed by adding couple of paragraphs in page: 4   line: 171-191

  1. How the dissemination of the research finding does could be more contributing?

This issue has been addressed by adding couple of paragraphs in page :  16line: 527-539

Reviewer 3 Report

Remote sensing

Paper title:     

Presence of the past: Digital narrative of the Dennys Lascelles 3 Concrete Wool Store; Geelong, Australia

Authors: Md Mizanur Rashid , Chin Koi Khoo, Sofija Kaljevic and Surabhi Pancholi

The paper covers an interesting topic of digital 3D modelling of architecture which has been used to explore and research the lost architectural asset of Dennys Lascelles Wool Store. The model has been set up with Rhino computer program from fragmented archival data available and photos of the original building. The key ideas are  1) “To initiate and demonstrate the process of understanding how digital place making can complement physical place making, and vice-versa, through utilization and reconstruction of collected heritage material” and 2) “To explore and establish a new conceptual framework that will define successful relationship between (a) recollection and (collective) memories and multiple narratives of the place, and (b) creating a web-based framework for bottom up user based and user driven storytelling online immersive environment”.

The paper logically develops the main idea thru examples, comparisons and literature review all the way from early wool times to reconstruction of the digital model based on the archival evidence. All information relevant to the building and its architecture was collated and a 3D virtual model of the building and the site was made based on the available data. The first part of the research is well written, persuasive and useful for the potential reader, but it is not scientifically new. The other part of the paper tries to demonstrate why such 3D model is useful, not only for commercial presentation, but also for the scientific research purposes. Here the paper could be more specific in making the valuable scientific contribution. For instance, it could be more specifically shown which are the advantages of such virtual model and which are those new revelations that cannot be gained from old pictures and historic material only. The authors are trying to show, that behind there is something more than a digital model, but this should be done more in scientific way, rather than enthusiastic one. The 4D model is also a scientific step in this direction, however it is not included/developed/explained enough in the paper. In my opinion, the second part of the paper which deals with the dissemination of the research findings, and will be the most valuable for the potential readers, could be improved and supplemented.

Author Response

Reviewer 3:

  1. Details of the 3D modelling process; Metric survey and confidence.

These issues has been addressed in page 6 , line 272-278 and page 8,  line 320-21.

  1. Expand and develop a more effective digital narrative;

This issue has been addressed by adding a paragraph in page 10 ;  line 348-353.

As well as adding an image explaining the urban transformation around the site in last 100 years. (Figure 9)

Round 2

Reviewer 1 Report

Dear Authors, I was not aware of this special issue of Remote Sensing, devoted to virtual modelling. If that is the case, than I don't see any reason why this paper should not be accepted, especially after those additions you have made according to other Reviewers' suggestions.